# Steering the reaction pathway of syngas-to-light olefins with coordination unsaturated sites of ZnGaO$_x$ spinel

Na Li[1,2,3,8], Yifeng Zhu[1,8], Feng Jiao [1,2,3,8], Xiulian Pan [1,2,3 ✉], Qike Jiang[2], Jun Cai[3,4,5], Yifan Li [1,6], Wei Tong [7], Changqi Xu[2], Shengcheng Qu[2], Bing Bai[1,3], Dengyun Miao[1,2,3], Zhi Liu[4,5] & Xinhe Bao [1,2 ✉]

Significant progress has been demonstrated in the development of bifunctional oxide-zeolite catalyst concept to tackle the selectivity challenge in syngas chemistry. Despite general recognition on the importance of defect sites of metal oxides for CO/H$_2$ activation, the actual structure and catalytic roles are far from being well understood. We demonstrate here that syngas conversion can be steered along a highly active and selective pathway towards light olefins via ketene-acetate (acetyl) intermediates by the surface with coordination unsaturated metal species, oxygen vacancies and zinc vacancies over ZnGaO$_x$ spinel−SAPO-34 composites. It gives 75.6% light-olefins selectivity and 49.5% CO conversion. By contrast, spinel−SAPO-34 containing only a small amount of oxygen vacancies and zinc vacancies gives only 14.9% light olefins selectivity at 6.6% CO conversion under the same condition. These findings reveal the importance to tailor the structure of metal oxides with coordination unsaturated metal sites/oxygen vacancies in selectivity control within the oxide-zeolite framework for syngas conversion and being anticipated also for CO$_2$ hydrogenation.

[1] State Key Laboratory of Catalysis, Dalian Institute of Chemical Physics, Chinese Academy of Sciences, 457 Zhongshan Road, Dalian 116023, PR China. [2] Dalian National Laboratory for Clean Energy, Dalian Institute of Chemical Physics, Chinese Academy of Sciences, 457 Zhongshan Road, Dalian 116023, PR China. [3] University of Chinese Academy of Sciences, 100049 Beijing, PR China. [4] State Key Laboratory of Functional Materials for Informatics, Shanghai Institute of Microsystem and Information Technology, Chinese Academy of Sciences, Shanghai 200050, PR China. [5] School of Physical Science and Technology, ShanghaiTech University, Shanghai 201210, PR China. [6] Department of Chemical Physics, University of Science and Technology of China, Jinzhai Road 96, Hefei 230026, PR China. [7] High Magnetic Field Laboratory, Hefei Institutes of Physical Science, Chinese Academy of Sciences, Hefei 230031, PR China. [8] These authors contributed equally: Na Li, Yifeng Zhu, Feng Jiao. ✉email: panxl@dicp.ac.cn; xhbao@dicp.ac.cn

Syngas is an important intermediate platform for the utilization of carbon resources such as coal, natural gas and biomass, which can be converted to a variety of high-value chemicals and fuels. The selectivity control in syngas chemistry remains a challenge although significant progress has been made in fundamental studies and industrial applications of Fischer-Tropsch synthesis (FTS) technology since its invention almost a century ago[1,2]. It was demonstrated that composite catalysts by coupling partially reducible metal oxides and zeolites or zeotypes (OXZEO) enabled syngas direct conversion to a variety of chemicals, e.g., light olefins, gasoline range isoparaffins, benzene-toluene-xylene (BTX), and even oxygenates, with their selectivities all surpassing the Anderson-Schultz-Flory (ASF) distribution limit[3–7]. For example, the selectivity of light olefins among hydrocarbons reached 80% at 17% CO conversion over $ZnCrO_x$-SAPO-34 at 400 °C, 2.5 MPa[3] while 49% CO conversion and 83% selectivity of light olefins over $ZnCrO_x$-AIPO-18 at 390 °C, 10 MPa[8]. $ZnCrO_x$-mordenite gave 83% ethylene selectivity and 7% CO conversion at 360 °C, 2.5 MPa[6]. Furthermore, similar metal oxide-zeolite systems were also developed for $CO_2$ hydrogenation to a variety of chemicals and fuels. For instance, $ZnZrO_x$-SAPO-34 for light olefins synthesis[9], $ZnZrO_x$-ZSM-5[10], and $ZnAlO_x$-H-ZSM-5[11] for aromatics synthesis, and $In_2O_3$-H-ZSM-5 for gasoline-range hydrocarbon synthesis[12].

It is generally recognized that within the framework of OXZEO catalyst concept, $CO/H_2$ activation takes place over metal oxides and C–C coupling over zeolites[3,4]. The partial reducibility of metal oxides was essential in controlling the overall activity of CO conversion. For example, partially reduced $MnO_x$ enabled CO dissociation and conversion to surface carbonate and carbon species, which were converted to $CO_2$ and hydrocarbons upon $H_2$ introduction. In comparison, no carbonate species were detected on unreduced $MnO_x$, revealing the pivotal role of surface oxygen vacancies in syngas conversion[13]. Similarly, the reducibility phenomenon was reported for Zn-based catalysts[13–16]. $ZnAl_2O_4$ with a Zn/Al ratio of 1/2 achieved the highest CO or $CO_2$ conversion, which was attributed to the largest amount of oxygen vacancies thus promoting CO and $CO_2$ activation and conversion[14]. Zn/Cr ratios also affected $H_2$ reduction and thus the formation ability of oxygen vacancies, which significantly influenced the activity and selectivity of syngas conversion[15,17]. $ZnO$-$ZrO_2$ aerogel catalyst provided high surface area and large amount of oxygen vacancies, which also played important roles in bifunctional $ZnO$-$ZrO_2$−ZSM-5 catalyst for $CO_2$ hydrogenation to aromatics[16]. Despite significant progress, the actual structure of oxygen vacancies, and their catalytic roles are far from being well understood in OXZEO catalysis.

Here, we show that the reaction pathways of syngas conversion strongly depend on the defect sites of metal oxides and hence the final product distribution. Partially reducible $ZnGaO_x$ spinel containing coordination unsaturated $Ga^{3+}$ sites and oxygen vacancies steers syngas conversion pathway towards light olefins, whereas spinel with a similar composition but only little oxygen vacancies and zinc vacancies is remarkably less active and non-selective for light olefins.

## Results

**Structure and catalytic performance of $ZnGaO_x$−SAPO-34 composites.** $ZnGaO_x$ oxides prepared by coprecipitation method were denoted as $ZnGaO_{x\_NP}$ and those by hydrothermal method were named as $ZnGaO_{x\_F}$. The transmission electron microscopy (TEM) and scanning electron microscopy (SEM) images in Fig. 1a–d, and Supplementary Figs. 1, 2a, b, and 3a–d show that the hydrothermal $ZnGaO_{x\_F}$ sample exhibits a hydrangea shape formed by thin flakes of ~21 nm thickness and micrometer size

on the plane direction. In comparison, all coprecipitation $ZnGaO_{x\_NP}$ samples exhibit nanoparticle morphologies (Fig. 1e–g, and Supplementary Figs. 2c, d, 3e–j, and 4), which do not appear to preferentially expose certain crystal faces. Various faces are observed including {111} and {010} in the [101] orientation over $ZnGaO_{x\_NP}$ sample (Fig. 1g and Supplementary Fig. 3e–j).

X-ray diffraction (XRD) patterns confirm that all $ZnGaO_x$ samples exhibit the same spinel crystal phase (JCPDS number of 38-1240, Fig. 1h, and Supplementary Figs. 5 and 6a) with the Zn/Ga ratio ranging from 0.2 to 3.1 (Supplementary Table 1). Therefore, they are non-stoichiometric spinels[18,19]. We first compared a nanoparticle spinel $ZnGaO_{x\_NP-A}$ with a nanoflake spinel $ZnGaO_{x\_F}$ with similar surface and bulk Zn/Ga molar ratios, as analyzed by inductively coupled plasma optical emission spectrometer (ICP-OES), scanning electron microscopy with energy dispersive X-ray detector (SEM-EDX), and X-ray photoelectron spectroscopy (XPS) (Fig. 2a, Supplementary Table 1). Interestingly, these two oxides give significantly different catalytic performance in syngas conversion upon being physically mixed with SAPO-34, respectively (Fig. 2b, Supplementary Table 2). CO conversion over $ZnGaO_{x\_NP-A}$−SAPO-34 is 32.3%, almost 5 times higher than 6.6% over $ZnGaO_{x\_F}$−SAPO-34, while the light olefins selectivity over the former is also 5 times higher than that over the latter (77.6% versus 14.9%). Note that hydrocarbons selectivity in this study is reported excluding $CO_2$ to simplify the discussion since $CO_2$ selectivity is similar for all catalysts (Supplementary Table 2). Even being normalized by the specific surface area of oxides, $ZnGaO_{x\_NP-A}$ still exhibits a yield of light olefins 7 times higher than $ZnGaO_{x\_F}$ (0.047 versus 0.007 mmol $m^{-2}$ $h^{-1}$) (Fig. 2c).

The catalytic activity can be further optimized by varying the composition of $ZnGaO_x$, as shown in Supplementary Tables 1 and 2. Notably, all nanoparticle $ZnGaO_{x\_NP}$ samples demonstrate much superior performance than $ZnGaO_{x\_F}$ upon being physically mixed with SAPO-34 (Fig. 2b, c and Supplementary Table 2). $ZnGaO_{x\_NP}$ with a surface Zn/Ga molar ratio of 1.2 (Fig. 2a) gives a highest CO conversion, i.e., 49.3% with 75.6% selectivity of light olefins (Fig. 2b and Supplementary Table 2). This CO conversion is higher than most results reported for Cr-free oxides under similar reaction conditions[20]. The formation activity of light olefins is 0.14 mmol $m^{-2}$ $h^{-1}$, which is 20 times higher than that over $ZnGaO_{x\_F}$ (Fig. 2c). Moreover, $ZnGaO_{x\_NP}$−SAPO-34 delivers a rather good stability. CO conversion remains at 46% and selectivity of light olefins at 70% after 120 h on stream (Supplementary Fig. 7).

To understand the distinctively different activity of nanoparticle and nanoflake $ZnGaO_x$ spinel, we first looked into the activity of $ZnGaO_{x\_NP}$ and $ZnGaO_{x\_F}$ alone in syngas conversion. The results in Supplementary Table 3 show that both oxides give similar CO conversion around 5.5%. However, they behave significantly different when they combined with SAPO-34 respectively as composites. $ZnGaO_{x\_NP}$−SAPO-34 provides CO conversion > 37% and light olefins selectivity > 71% at different OX/ZEO mass ratios (Supplementary Fig. 8a). It indicates that the reaction equilibrium is successfully shifted and the reaction channel from intermediates to light olefins is opened up in the presence of SAPO-34, consistent with a recent theoretical study[21]. Thus, it forms a tandem catalytic process. Furthermore, the composite with OX/ZEO = 1 (mass ratio) gives optimal performance. By contrast, introducing SAPO-34 to $ZnGaO_{x\_F}$ hardly affects the overall conversion (<8%) and light olefins selectivity (<16%) with the OX/ZEO ratio changing from 1/2 to 2/1 (Supplementary Fig. 8b, c). The main products are methane (45%) and light paraffins (40%). It implies that $ZnGaO_{x\_F}$ is intrinsically of low activity or the intermediates generated over $ZnGaO_{x\_F}$ cannot be effectively

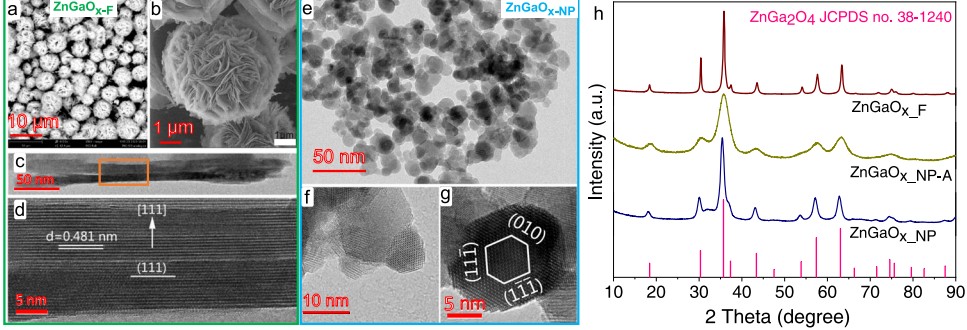

**Fig. 1 ZnGaO$_x$ structure. a-d** ZnGaO$_{x\_F}$ sample. **a, b** Scanning electron microscopy (SEM) images. **c, d** High-resolution transmission electron microscopy (HRTEM) side view images. **d** The enlarged image of a selected area with orange frame in **c** viewed along the [1$\bar{1}$0] orientation. **e-g** ZnGaO$_{x\_NP}$ sample. TEM images with **g** viewed along the [101] orientation. Note that [uvw] indexed a crystal axis, (hkl) a crystal plane, and {hkl} a group of crystal planes with the same atomic configuration[70]. **h** XRD pattern of ZnGaO$_x$ samples.

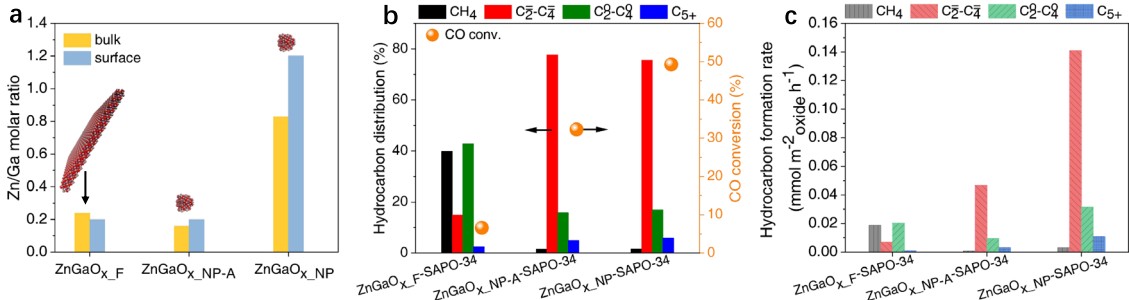

**Fig. 2 Elemental ratio and reaction performance. a** Zn/Ga molar ratio of ZnGaO$_x$ samples corresponding to the data in Supplementary Table 1. The inset is morphology diagram, with purple, dark pink, and red balls referring to Zn, Ga, and O atoms, respectively. **b** Reaction performance of syngas conversion over ZnGaO$_x$−SAPO-34. **c** Hydrocarbon formation rate normalized by specific surface area of oxides. Reaction conditions: OX/ZEO = 1 (mass ratio, 20–40 mesh), H$_2$/CO = 2.5 (v/v), 400 °C, 4 MPa, 1600 mL g$^{-1}$ h$^{-1}$.

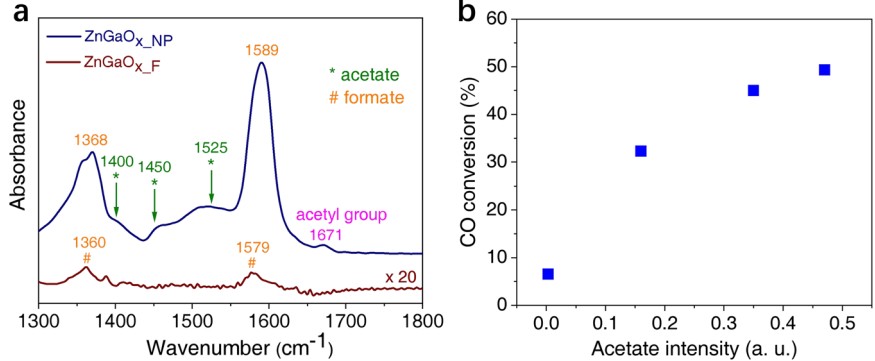

**Fig. 3 Surface intermediates over ZnGaO$_x$ oxides and the relationship with catalytic performance. a** In-situ FT-IR differential spectra of syngas conversion over H$_2$-reduced ZnGaO$_{x\_NP}$ (navy line) and ZnGaO$_{x\_F}$ (brown line) at 400 °C. **b** CO conversion as a function of acetate intensity at 1525 cm$^{-1}$ of FT-IR spectra of different ZnGaO$_x$ samples in Supplementary Fig. 9a.

converted to desired products by SAPO-34 catalyst in contrast to ZnGaO$_{x\_NP}$. The reaction most likely has gone through different pathways over the two types of oxides.

**In-situ FT-IR analysis of reaction intermediates over ZnGaO$_x$ spinel.** The in-situ Fourier Transform Infrared (FT-IR) spectra in Fig. 3a confirm the distinct intermediates generated over the two ZnGaO$_x$ oxides. Upon exposing ZnGaO$_{x\_NP}$ to syngas at 400 °C, two strong absorption bands appear at ~1368 and ~1589 cm$^{-1}$, which are generally ascribed to formate species[22]. Such formate species is also observed over ZnGaO$_{x\_F}$, although the intensity is weaker. Formate species has been widely observed over methanol

synthesis catalysts and is generally considered to be the precursor of methanol[4,9,23,24]. It was also reported for the metal oxides of the OXZEO composites which go via methanol-olefins pathway[4,8,25,26]. However, neither the intensity of IR formate signal correlates well with CO conversion of the corresponding OXZEO catalysts (Supplementary Fig. 9) nor the methanol concentration produced by ZnGaO$_x$ correlates with the hydrocarbons produced by OXZEO catalysts (Supplementary Fig. 10).

In comparison, over ZnGaO$_{x\_NP}$ surface, additional signals at ~1400 ($\delta_{CH3}$), ~1450 ($\upsilon_{C-O}$) and ~1525 cm$^{-1}$ ($\upsilon_{C=O}$) are obviously observed (Fig. 3a and Supplementary Fig. 9a), which are characteristic acetate species[27–30]. The band at 1671 cm$^{-1}$ is assigned to acetyl group[27–30]. The acetate species likely originates

from ketene being chemisorbed on the surface hydroxyl group, and acetyl group could be the product of hydrogenated ketene in the absence of zeotypes. Acetate species, a product of C–O breaking and C–C coupling, was not reported over typical methanol synthesis catalysts previously[4,9,23,24]. Formation of acetate species over ZnGaO$_{x\text{-NP}}$ was also validated by solid-state Nuclear Magnetic Resonance study[31]. Figure 3b displays that CO conversion of ZnGaO$_x$−SAPO-34 bifunctional catalysts correlates well with the intensity of the representative IR signal for acetate species (1525 cm$^{-1}$). Therefore, the reaction likely proceeds through ketene/acetyl/acetate pathway over ZnGaO$_x$-SAPO-34 although methanol contribution cannot be completely excluded because SAPO-34 is a classical catalyst for methanol-to-olefins.

**Unraveling the coordination unsaturated metal sites over ZnGaO$_x$ spinel.** To understand the origin of different reaction pathways over the two types of ZnGaO$_x$ spinel, we set out to investigate their structures and active sites. Although metal oxides are less studied as catalysts directly, the increasing number of studies have proposed the important role of oxygen vacancies[3,25,26] in OXZEO catalyzed syngas conversion. Therefore, we turned to CO-temperature programmed reduction (TPR) first. The profiles in Fig. 4a demonstrate a much more facile reduction of ZnGaO$_{x\_NP}$ than ZnGaO$_{x\_F}$. A strong signal of CO$_2$ centers around 300 °C over ZnGaO$_{x\_NP}$. This signal appears to overlap with another one starting from ~ 400 °C, which is likely CO disproportionation. Nevertheless, the integrated area of CO$_2$ signals below 400 °C in CO-TPR (Fig. 4a, Supplementary Fig. 11, and Supplementary Tables 4 and 5) can still reflect the reducibility. In contrast, the reduction peak is significantly weaker over ZnGaO$_{x\_F}$ indicating very few reducible defect sites below 400 °C. H$_2$-TPR shows a similar trend (Supplementary Fig. 12). Figure 4b

displays that the specific formation rates of hydrocarbons and light olefins correlate monotonically with the reducibility of ZnGaO$_x$ oxides regardless of nanoparticles or nanoflakes, which is consistent with previous studies[3,13,25,26], revealing again the essential role of reducibility.

The reduction process can remove surface oxygen atoms, thereby leaving oxygen vacancies and coordinatively unsaturated metal sites, which are further investigated by *quasi*-in-situ electron paramagnetic resonance (EPR)[32–35]. Figure 4c shows that fresh ZnGaO$_{x\_NP}$ exhibits a very weak EPR signal at g = 2.004. However, it intensifies significantly together with a new signal showing up at g = 1.97 upon H$_2$ reduction (Supplementary Fig. 13a). In comparison, only a weak signal attributed to manganese impurity[36] is detected for fresh ZnGaO$_{x\_F}$ (Fig. 4d, and Supplementary Fig. 13b), which may has been brought in during catalyst synthesis (Supplementary Table 1). Upon H$_2$ reduction, a signal at g = 2.004 appears and no other signals are observed over nanoflakes. g = 2.004 signal was frequently reported to be related to the presence of a free electron in the conduction band[37,38], or defect sites over singly ionized zinc vacancy of zinc-based oxides[32,39–42]. However, it was also assigned to unpaired electron trapped at an oxygen vacancy site of oxides such as ZnO[35,41,43], Zn$_x$Ga$_y$O$_z$[37,44,45], or TiO$_2$[38,46]. The assignment of g = 1.97 also remains controversial in different studies, e.g., singly ionized oxygen vacancies with one trapped electron[33,47], zinc vacancies[42,48,49], or donor centers such as ionized impurity atoms in the crystal lattice of ZnO oxide[50,51]. Volatility of zinc species has been frequently reported for ZnO and Zn-based oxides, which is facilitated in vacuum and hydrogen atmosphere[52–55]. Thus formation of zinc vacancies over both reduced ZnGaO$_x$ can be expected upon H$_2$ reduction due to removal of oxygen atoms. This is confirmed by in-situ ambient pressure X-ray photoelectron spectroscopy (AP-XPS) experiments (the insets of Fig. 4c, d and Supplementary Table 6).

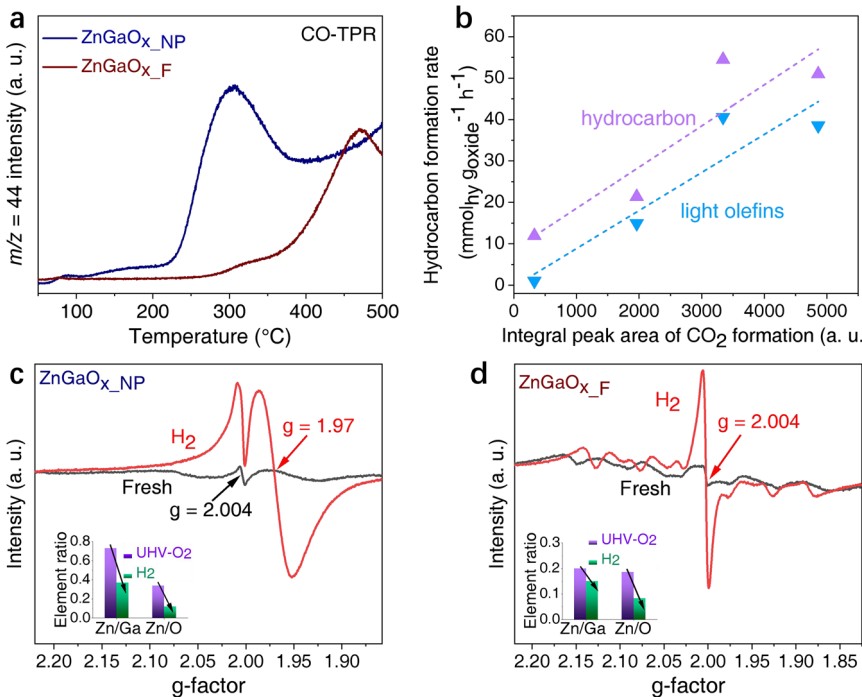

**Fig. 4 Structural analysis of ZnGaO$_x$ by TPR, EPR. a** CO-TPR profiles with *m/z* = 44 (CO$_2$) signals in the effluents monitored by an online mass spectrometer. **b** Mass specific hydrocarbon formation rate as a function of integral area of CO$_2$ signals below 400 °C in CO-TPR profiles of different ZnGaO$_x$. Reaction conditions: OX/ZEO = 1/4 (mass ratio), H$_2$/CO = 2.5 (v/v), 400 °C, 4 MPa, and 20,000 mL g$^{-1}$ h$^{-1}$. **c, d** *Quasi*-in-situ EPR spectra of ZnGaO$_x$ before and after H$_2$ reduction, and the inset showing the surface Zn/Ga and Zn/O ratios determined by AP-XPS results. The purple and green colors refer to treatment conditions of UHV-O$_2$ (degassed in ultra-high vacuum, and then exposed to O$_2$) and H$_2$, respectively. **c** ZnGaO$_{x\_NP}$. **d** ZnGaO$_{x\_F}$.

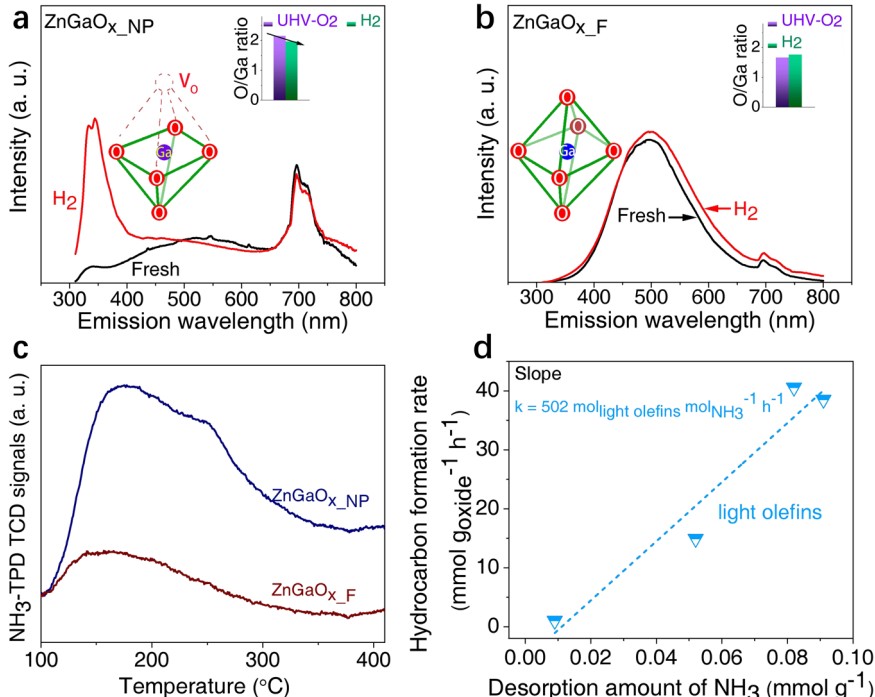

**Fig. 5 Structural analysis of ZnGaOₓ by photoluminescence and NH₃-TPD. a, b** *Quasi*-in-situ Photoluminescence emission spectroscopy. H₂-reduced samples in comparison to the fresh ones at the excitation wavelength of 290 nm (Supplementary Fig. 14), with the inset showing the surface ratio of O/Ga measured by in-situ AP-XPS. The inset model showing the Ga-O octahedral structure after H₂ treatment. **a** ZnGaOₓ_NP. **b** ZnGaOₓ_F. **c** NH₃-TPD profiles of H₂-reduced ZnGaOₓ oxides. **d** Mass specific hydrocarbon formation rate as a function of the amount of medium strength acid sites of ZnGaOₓ estimated by NH₃-TPD. Reaction conditions: OX/ZEO = 1/4 (mass ratio), H₂/CO = 2.5 (v/v), 400 °C, 4 MPa, 20,000 mL g⁻¹ h⁻¹.

Therefore, it is reasonable to attribute the $g = 2.004$ signal in both H₂-reduced oxides to singly ionized zinc vacancies. Since the reducibility of ZnGaOₓ_F at 400 °C is relatively low (Fig. 4a and Supplementary Fig. 12), the signal intensity of oxygen vacancies over ZnGaOₓ_F should also be low, in contrast to significant reduction signal over ZnGaOₓ_NP. Therefore, the $g = 1.97$ signal of H₂-reduced ZnGaOₓ_NP can be attributed to singly ionized oxygen vacancies.

Photoluminescence (PL) spectroscopy was conducted to further elucidate the defect structures of ZnGaOₓ oxides. Figure 5a shows emission peaks around 700 nm over both the fresh and reduced ZnGaOₓ_NP, which are generally attributed to oxygen vacancies[56,57], in agreement with CO-TPR and EPR results. In addition, upon H₂ reduction, another emission signal near 350 nm in the ultraviolet (UV) region becomes significantly intensified, consistent with the previous observation for ZnGa₂O₄[58]. This was attributed to the formation of the distorted Ga-O octahedral structure, due to the removal of O atoms and thus forming coordination unsaturated Ga³⁺ sites, as displayed in the structure model in Fig. 5a[56,59–61]. This is further validated by a decreased O/Ga ratio of reduced ZnGaOₓ₋NP by AP-XPS (the inset of Fig. 5a), whereas ZnGaOₓ_F does not show much change of the surface O/Ga ratio (the inset of Fig. 5b). In comparison, the fresh ZnGaOₓ_F exhibits an emission spectrum significantly different from that of ZnGaOₓ_NP, with a strong and wide signal at 400~650 nm and a weak signal around 700 nm (Fig. 5b). The former signal generally corresponds to the characteristic coordination saturated Ga³⁺ in the Ga-O octahedral structure (model of Fig. 5b)[61,62] due to the charge transfer between Ga³⁺ ions located at the center of octahedral sites and its six first-neighbor O²⁻ ions[56,61]. Furthermore, the *quasi*-in-situ H₂-treated ZnGaOₓ_F at 400 °C gives almost the identical spectrum as the fresh sample, indicating no obvious electronic structure change (Fig. 5b).

The above results indicate that the coordination unsaturated Ga³⁺ sites, oxygen vacancies and zinc vacancies co-exist on reduced ZnGaOₓ_NP oxide, while only oxygen vacancies and zinc vacancies exist on reduced ZnGaOₓ_F. The coordinatively unsaturated Ga³⁺ sites generally exhibit Lewis acidity[63,64], which is evidenced by in-situ FT-IR differential spectra of pyridine adsorption on the surface of ZnGaOₓ oxides (Supplementary Fig. 15). The amount of Lewis acid sites can be quantified by temperature-programmed desorption of ammonia (NH₃-TPD)[26,65]. Figure 5c and Supplementary Fig. 16a show that all reduced ZnGaOₓ samples give broad and asymmetric NH₃ desorption peaks in the range of 100~400 °C, but the concentration differs (Supplementary Table 7). Fitting of NH₃-TPD profiles (Supplementary Fig. 16b–e and Supplementary Table 8) indicates the presence of weak (around 170 °C) and medium strength acid sites (around 250 °C)[66]. The NH₃-desorption peak below 200 °C is generally related to hydrogen-bonded physisorption sites[67], while the peak around 250 °C could be contributed by the defect sites of coordination unsaturated sites[26,65]. Therefore, the number of defect sites on ZnGaOₓ surface could be quantified by the amount of NH₃ desorbing from the medium strength acid sites. Interestingly, as shown in Fig. 5d, the mass specific light olefins formation rate is positively correlated with the concentration of these coordination unsaturated metal sites, but not with that of weak strength acid sites (Supplementary Fig. 17). Thus, the light olefins formation rate per defect site can be estimated to be 502 h⁻¹, assuming one defect site adsorbing one NH₃ molecule. The above results demonstrate that the presence of coordination unsaturated Ga³⁺ together with oxygen vacancies and zinc vacancies lead to a much more active ZnGaOₓ spinel in generating ketene-acetate (acetyl) intermediates. Interestingly, Lai et al. recently also revealed the essential role of coordination unsaturated Cr³⁺ together with the oxygen vacancies in the cleavage of the C–O bond over the highly reduced ZnCr₂O₄ (110)

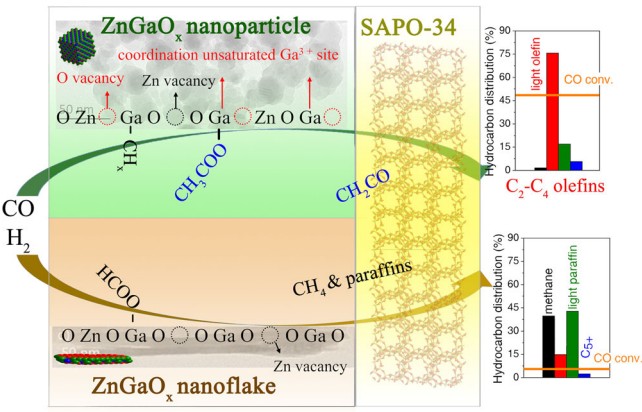

**Fig. 6 Reaction pathways over ZnGaO$_x$-SAPO-34 composites.** ZnGaO$_x$ oxides containing coordination unsaturated Ga$^{3+}$, oxygen vacancy and zinc vacancy sites are much more active and selective in syngas conversion to light olefins.

surface[21]. Thus, incorporation of SAPO-34 would direct the reaction pathway towards light olefins (Fig. 6).

## Discussion

ZnGaO$_x$ spinels with similar compositions but different morphologies were synthesized, which allows to elucidate the catalytic role of different defect sites over metal oxides in OXZEO catalyzed syngas conversion. *Quasi*-in-situ PL, EPR and in-situ FT-IR reveal that ZnGaO$_{x\_NP}$ (nanoparticles) is much more reducible, which gives the coordination unsaturated Ga$^{3+}$ species, oxygen vacancies and zinc vacancies. Such a surface facilitates the formation of ketene-acetate (acetyl) intermediates during CO/H$_2$ activation, which allows subsequent conversion to light olefins by SAPO-34 and displacement of the reaction equilibrium. Consequently, the selectivity of light olefins reaches 75.6% at 49.3% CO conversion, which is 9 times higher than that obtained by ZnGaO$_{x\_NP}$ alone. In comparison, ZnGaO$_{x\_F}$ (nanoflakes) containing only oxygen and zinc vacancies catalyzes CO/H$_2$ activation generating formate species, which are hardly converted to light olefins by SAPO-34 because CO conversion is only 6.6% and light olefins selectivity is only 14.9%. Instead, the products are mainly composed of CH$_4$ and paraffins for both ZnGaO$_{x\_F}$ alone and ZnGaO$_{x\_F}$−SAPO-34 composite. Although the detailed structure of the active sites and the elementary steps forming intermediates still need further investigation, the results here have already demonstrated that the structure of reducible metal oxides can be tailored to convert syngas selectively to value-added chemicals. These findings are also expected to be applicable to CO$_2$ hydrogenation to value-added chemicals and fuels.

## Methods

**Catalyst preparation**. ZnGaO$_{x\_NP}$, where NP denoted as nanoparticles, was synthesized by a coprecipitation method with the temperature of water bath kept at 60~65 °C and pH kept at 9~10. Aqueous solutions of Zn(NO$_3$)$_2$·6H$_2$O and Ga(NO$_3$)$_3$·nH$_2$O were prepared as the precursors and their molar ratios were 1:10, 1:6, 1:4, and 1:2, respectively. An aqueous solution of NaOH and Na$_2$CO$_3$ with a molar ratio of 7.1:1 was used as the precipitant. After precipitation, the suspensions were aged for 2 h under continuous stirring. The precipitates were washed with water, dried at 60 °C and then 110 °C overnight, followed by calcination at 500 °C for 1 h in air, respectively. The resulting samples were named as ZnGaO$_{x\_NP-A}$, ZnGaO$_{x\_NP-B}$, ZnGaO$_{x\_NP-C}$, and ZnGaO$_{x\_NP}$, respectively. The effect of calcination temperature, $T$, was also studied in the range of 600~800 °C for ZnGaO$_{x\_NP}$ (named as ZnGaO$_{x\_NPT}$).

ZnGaO$_{x\_F}$, where F denoted as nanoflakes, was prepared by a hydrothermal method by adapting the previously reported method[68]. In detail, 1.21 g 2H$_2$O · Zn(CH$_3$COO)$_2$ and 2.81 g nH$_2$O · Ga(NO$_3$)$_3$ were dissolved in a mixed solution of 110 mL water and 55 mL ethylenediamine. After continuous stirring at room temperature for 1 h, the mixed solution was transferred into a 200 mL

hydrothermal kettle, and heated at 180 °C for 24 h. The product was separated by centrifugation, washed several times with water, and then dried overnight at 60 °C, followed by calcination at 500 °C for 1 h in air.

SAPO-34 was synthesized following a hydrothermal method similar to a previous report[3]. Typically, 30% silica sol, AlOOH, 85% phosphoric acid and triethylamine (TEA) were well dispersed in distilled water with a mass ratio of SiO$_2$:Al$_2$O$_3$:H$_3$PO$_4$:TEA:H$_2$O = 0.11:1:1.8:3.4:10. Then the mixture was placed in a Teflon-lined autoclave, and kept at 200 °C for 24 h. The resulting solid product was collected by centrifuging and washed with water till the pH of the supernate was 7.0–7.5. After drying for over 12 h at 110 °C, the white powder was calcined at 550 °C for 4 h in air with a heating rate of 1 °C/min.

**Catalyst characterization**. X-ray diffraction (XRD) was measured on a PANalytical Empyrean-100 equipped with a Cu K$_\alpha$ radiation source ($\lambda$ = 1.5418 Å), operated at 40 mA and 40 kV. XRD patterns were recorded in the range of 2 theta = 10~90°. The crystal size was estimated using the Scherrer equation. Nitrogen adsorption−desorption experiments were carried out on a Quantachrome NOVA 4200e instrument. Before analysis, samples were degassed under vacuum at 300 °C for 5 h. Isotherms were recorded at 77 K. A non-local density function theory (NLDFT) pore size method was used. High-resolution transmission electron microscopy (HRTEM) images were obtained using a JEOL JEM-2100 electron microscope operated at an accelerating voltage of 200 kV. Before tests, the samples were ultrasonically dispersed in ethanol and a drop of the solution was placed onto a copper grid coated with a thin microgrids support film. High-resolution scanning electron microscopy (HRSEM) images were obtained using a Carl Zeiss Orion NanoFab Helium ion microscope. The low-resolution scanning electron microscopy (SEM) tests were performed on a Phenom proX apparatus with energy dispersive X-ray detector (EDX) elemental analysis. The accelerating voltage was 15 kV. The elemental content was measured using Inductively Coupled Plasma Optical Emission Spectrometer (ICP-OES). Samples were dissolved in aqua regia solution and then sealed in an autoclave with Teflon lining. The autoclave was then placed in a microwave reactor for 0.5 h. The samples were then measured on a PerkinElmer ICP-OES 7300DV apparatus. X-ray photoelectron spectroscopy (XPS) spectra were recorded on a SPECS PHOIBOS-100 spectrometer using an Al K$_\alpha$ (hν = 1486.6 eV, 1 eV = 1.603 × 10$^{-19}$ J) X-ray source. The Ga 2p$_{3/2}$ binding energy at 1118.7 eV was used for calibration. Typically, only the Ga 2p$_{3/2}$ component of the Ga 2p and Zn 2p$_{3/2}$ of the Zn 2p regions are fitted and quantified. The atomic ratio of elements $i$ to $j$ ($n_i/n_j$) on the oxide surface was calculated based on

$$\frac{n_i}{n_j} = \frac{I_i}{S_i} \div \frac{I_j}{S_j} \tag{1}$$

Where $I$ represented the area of the characteristic peak and $S$ represented the atomic sensitivity factor in Eq. (1), which was referred to the previous literature[69]. Ambient Pressure X-ray Photoelectron Spectroscopy (AP-XPS) experiments were carried out on SPECS PHOIBOS-150 ambient pressure XPS with Al K$_\alpha$ as the X-ray source. ZnGaO$_x$ oxides were first degassed in ultra-high vacuum (UHV) at 400 °C, and then heated in 0.5 mbar O$_2$. After pretreatment, 0.5 mbar H$_2$ was introduced into the analysis chamber. The Ga 2p$_{3/2}$ binding energy at 1118.7 eV was used for calibration. Temperature-programmed desorption of NH$_3$ (NH$_3$-TPD) was performed on a Micromeritics AutoChem 2910 instrument equipped with a thermal conductivity detector (TCD). Typically, 100 mg sample was loaded into a U-shape reactor. Before NH$_3$-TPD experiment, sample was pretreated at 400 °C for 1 h under flowing H$_2$ and then heated under flowing Ar at 500 °C for 1.5 h. After cooling down to 100 °C under flowing Ar, the sample was exposed to 5 vol.% NH$_3$/He. Then, the sample was swept by Ar at the same temperature until a stable baseline was obtained. Subsequently, the signal was recorded while the temperature increased from 100 to 600 °C at a heating rate of 10 °C/min. Temperature-programmed reduction (TPR) was performed on another Micromeritics AutoChem 2910 instrument equipped with a TCD. Typically, 100 mg sample was loaded into a U-shape reactor. Before CO-TPR or H$_2$-TPR experiment, sample was pretreated at 500 °C for 1 h in flowing Ar. After cooling down to room temperature under flowing Ar, the TPR profile was recorded in 5 vol.% CO/He or 1 vol.% H$_2$/Ar at a heating rate of 10 °C/min with the effluents monitored by an online quadrupole mass spectrometer (MS). Electron paramagnetic resonance (EPR) spectra were collected at 7 K on a Bruker A200 EPR spectrometer operated at the X-band frequency using power 1.0 mW, modulation amplitude 4.00 G and receiver gain 10000. The photoluminescence (PL) spectra were measured using QM400 with a Xe-lamp as the excitation source at room temperature. The excitation wavelength was fixed at 290 nm. H$_2$ reduction was conducted at 400 °C for 1 h, and then sealed for *quasi*-in-situ study of EPR and PL. In-situ Fourier Transform Infrared (FT-IR) transmission spectra were recorded on a BRUKER INVENIO S spectrometer equipped with a quatz cell. Before tests, sample was pretreated in H$_2$ atmosphere at 450 °C for 1 h. After cooling down to room temperature, the background spectrum was recorded. Then, sample was exposed to syngas atmosphere at 1 atm and heated to 400 °C for 1 h. After cooling down to room tempetature, sample spectrum was recorded. Each spectrum was obtained by averaging 32 scans collected at 4 cm$^{-1}$ resolution. The sample spectrum was subtracted by the background spectrum. Pyridine adsorption test was conducted using the same facilities. The sample was degassed in vacuum at 450 °C

for 1 h. Then pyridine was introduced at room temperature and subsequently degassed by evacuation. All spectra were collected under room temperature.

**Catalytic reaction tests**. Catalytic reaction was performed in a continuous flow, fixed-bed stainless steel reactor equipped with a quartz lining. Typically, 300 mg composite catalyst (20–40 meshes) with oxide/zeolite = 1/1 (mass ratio) was used. 5 vol.% Ar was added to syngas as the internal standard for online gas chromatography (GC) analysis. Reaction was carried out under conditions: $H_2/CO$ = 2.5 (v/v), 400 °C, 4.0 MPa, gas hourly space velocity (GHSV) = 1600 mL $g^{-1}$ $h^{-1}$ unless otherwise stated. Products were analyzed by an online GC (Agilent 7890B), equipped with a TCD and a flame ionization detector (FID). Hayesep Q and 5 A molecular sieves packed columns were connected to TCD whereas HP-FFAP and HP-AL/S capillary columns were connected to FID. Oxygenates and hydrocarbons up to $C_{12}$ were analyzed by FID, while CO, $CO_2$, $CH_4$, $C_2H_4$, and $C_2H_6$ were analyzed by TCD. $CH_4$, $C_2H_4$, and $C_2H_6$ were taken as a reference bridge between FID and TCD.

CO conversion ($Conv_{CO}$) was calculated on a carbon atom basis, i.e.

$$Conv_{CO} = \frac{CO_{inlet} - CO_{outlet}}{CO_{inlet}} \times 100\% \qquad (2)$$

where $CO_{inlet}$ and $CO_{outlet}$ in Eq. (2) represented moles of CO at the inlet and outlet, respectively.

$CO_2$ selectivity ($Sel_{CO_2}$) was calculated according to

$$Sel_{CO_2} = \frac{CO_{2outlet}}{CO_{inlet} - CO_{outlet}} \times 100\% \qquad (3)$$

where $CO_{2outlet}$ in Eq. (3) denoted moles of $CO_2$ at the outlet.

The selectivity of individual hydrocarbon $C_nH_m$ ($Sel_{C_nH_m}$) among hydrocarbons (free of $CO_2$) in Eq. (4) was calculated according to

$$Sel_{C_nH_m} = \frac{nC_nH_{m_{outlet}}}{\sum_1^n nC_nH_{m_{outlet}}} \times 100\% \qquad (4)$$

Little $C_{12+}$ hydrocarbons were detected. The selectivity to oxygenates was below 1%C and therefore neglected. The carbon balance over the OXZEO catalysts was over 95%.

## Data availability
All data supporting the findings of this study are available within the paper and its supplementary information files.

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

## Acknowledgements

This work was supported by the Ministry of Science and Technology of China (No. 2018YFA0704503, F.J.), the Chinese Academy of Sciences (XDA21020400, X.P.), the National Natural Science Foundation of China (Grant Nos. 91945302, X.P.; 22002153, F.J.; 22008234, D.M.), the Youth Innovation Promotion Association of Chinese Academy of Sciences (2019184, F.J.), Dalian Science and Technology Innovation Fund (2020JJ26GX028, F.J.), the Natural Science Foundation of Liaoning (2020-BS-019, F.J.). The lab-based SPECS AP-XPS instrument was supported by $ME^2$ project under contract no. 11227902 (Z.L.) from National Natural Science Foundation of China. We thank the High Magnetic Field Laboratory of the Chinese Academy of Sciences for the EPR measurement, and Professors Shengfa Ye and Jihu Su for discussion on EPR results, Professor Xueqing Gong and Fei Li for discussion on photoluminescence results.

## Author contributions

N.L., Y.Z., and F.J. contributed equally to this work. N.L. performed most material synthesis, characterization and catalytic activity tests. Y.Z. performed early experimental explorations. F.J. designed (quasi-) in-situ reactors and participated in most characterization as well as data analysis. X.P. and X.B. initiated the project. X.P., F.J. and Y.Z. designed the experiments. Q.J. carried out some HRTEM characterization and corresponding analysis. J.C. and Y.L. performed XPS measurements. W.T. participated in part EPR exploratory experiments and analysis. C.X. and S.Q. participated in samples synthesis, catalytic activity tests and characterization. B.B. assisted in IR test. D.M. participated in the discussion of the results. Z.L. participated in APXPS analysis. N.L., F.J., X.B. and X.P. wrote the manuscript. All authors discussed the manuscript and have given approval to the final version of the manuscript.

## Competing interests

The authors declare no competing interests.
