## [Peer Review File · Nature Communications]

Title: Steering the reaction pathway of syngas-to-light olefins with coordination unsaturated sites of ZnGaOx spinelREVIEWER COMMENTS

Reviewer #1 (Remarks to the Author):

Tandem catalysis is a key aspect in C1 chemistry and catalysis, for the conversion of CO, CO₂, methanol, methane, etc. Despite general recognition of the importance of defect sites (catalytic sites) on metal oxides including spinel in syngas (CO+H₂) conversion, there still exists less knowledge about the active sites on oxides for CO/H₂ activation and debate on the reaction routes because of its complexity.

This paper discloses deep insights on the defect sites of ZnGaOx spinel. It is an interesting and timely study demonstrating the essential and intrinsic role of the expected types of defects, which can catalyze the reaction toward valuable chemicals such as olefins. The conclusion is convincing and well supported by multiple-aspect experimental evidences from a variety of characterizations including high resolution TEM, in situ FTIR, quasi-in-situ photoluminescence and EPR, besides AP-XPS, CO-TPR and NH₃-TPD. Therefore, I believe that this study deserves publication at Nature Communications. But I have several questions before acceptance.

a) Concerning this bifunctional catalyst for a consecutive reaction, how is the effect of the relative ratio of metal oxides and zeolites? Is there any limitation at the activity of the second step, zeolite catalysis, for ZnGaOx_F? I suggest the authors give more discussions on kinetics balance.

b) More detailed analysis and interpretation of the EPR data, related to the reaction direction, is expected.

c) Figure S5 displays a new diffraction peak at 2theta 32 degree. But it is not discussed. Is this peak from ZnO? or from the interface of ZnO and ZnGa₂O₄ forming over the catalyst? Does it also contribute to the reaction?

d) References format need to be normalized.

Reviewer #2 (Remarks to the Author):

“Steering the reaction pathway of syngas-to-light olefins with coordination unsaturated sites of ZnGaOx spinel” by Na Li, Yifeng Zhu, Feng Jiao, Xiulian Pan, Qike Jiang, Jun Cai, Yifan Li, Wei Tong, Changqi Xu, Shengcheng Qu, Bing Bai, Dengyun Miao, Zhi Liu, and Xinhe Bao.

The main conclusions in the manuscript are:

1) That syngas conversion can be steered towards light olefins via ketene-acetate (acetyl) intermediates by the coordination unsaturated Ga³⁺ species and oxygen vacancies over ZnGaOx spinel–SAPO-34 composites.

2) Spinel containing only zinc vacancies leads to formation of formate intermediate species, but only as spectators, which can hardly be converted to light olefins by subsequent SAPO-34 catalysis.

As discussed below, the conclusion that the formate surface intermediates are not important does not appear to be well founded based on data. Formate species are generally believed to be precursors for methanol synthesis and there are significant differences between the two main samples (ZnGaOx_NP

and ZnGaOx_F) of this study in the selectivity and activity to methanol. Methanol can subsequently be converted into olefins over SAPO-34. Consequently, the methanol selectivity and activity look as a straightforward explanation of the OXZEO data presented here. Furthermore, the quality of the manuscript could be improved by amending a number of imprecise statements/language, not sufficiently defined methods, considerations later turning into conclusions etc. In general I find that the manuscript does not contain the novelty and quality needed to merit publication in Nature Communication.

Specific comments:

Line 32-33: "SAPO-34 catalysis" I would say this is a phrase from a spoken language. Please re-phrase.

Line 46-49: Please also report conversion levels. This is very important information, when comparing selectivity of these reactions.

Line 51: "ZnGaO-SAPO-34" should it be "ZnGaOx-SAPO-34"

Figure 1: Please provide scalebars for all parts of this figure. In the part of the figure text that covers (a,b), it is not stated what material the SEM images show.

Line 83-86: The type of samples is not introduced before e.g. "Hydrothermal sample" is mentioned. The paper should contain a small intro paragraph with an introduction of the samples.

Figure 2a: I presume that the XPS values represent measurements of the oxidic samples?

Line 109-111: "...than the flake one ..." represents spoken language and this sentence should be carefully rewritten. "ZnGaOx_NP-B with a surface Zn/Ga molar ratio of 1.2" In Table S1, the Zn/Ga molar ratio of this sample is 0.3.

Line 116-117: Even though this study is about using ZnGaOx samples together with SAPO-34 for the OXZEO process, the purpose of this study is to unravel the different reactivities of dissimilar surface sites on the ZnGaOx particles. This calls for a thorough investigation also of the catalytic properties of the ZnGaOx samples themselves and this has only been done for two samples in this work (Table S3). This should be done for all ZnGaOx samples studied here because these studies seem very informative as the products observed are not processed further on the active sites of SAPO-34 and in that way more complexed to analyze and the ZnGaOx differences somewhat blurred. Taking the available activity data for the two tested pure ZnGaOx samples, it seems to me that they explain the data with ZnGaOx mixed with SAPO-34 well and provide an explanation that is different from that put forward by the authors: The ZnGaOx-NP sample is quite selective towards methanol and my rough estimate suggests that methanol is (at least) in equilibrium (please provide information in Table S3 of how close the methanol is to equilibrium), while this is not the case for ZnGaOx-F, which also forms more methane. Since methanol is at equilibrium for the ZnGaOx-NP sample the selectivity towards methanol is underestimated if the data in Table S3 is used to estimate the selectivity to methanol directly. If this sample is mixed with SAPO-34, the methanol is converted to olefins, so the data in Table S3 seems to show that the difference between the ZnGaOx-F and the ZnGaOx-NP samples is just the methanol selectivity and activity. Please comment.

Line 119-120: "CO conversion is enhanced remarkably over ZnGaOx_NP-SAPO-34 (Figure S6a), indicating a shifted reaction equilibrium by the tandem catalysis of SAPO-34" This appears to be correct, but the data in Table S3 strongly suggests that it is the methanol, which is (known to be) converted to olefins over SAPO-34.

Line 121-123: "By contrast, introducing SAPO-34 to ZnGaOx_F hardly affects the overall conversion (< 8%) and light olefin selectivity < 16% at a wide range of OX/ZEO ratios (Figure S6b)." This is not surprising considering the above analysis based on methanol as the precursor for olefins.

Figure 3: It seems that the sensitivity of the In-situ FTIR is not large enough for the ZnGaOx_F sample to judge whether the formate to acetyl ratio is the same as that for ZnGaOx_NP.

Line 134-136: "They (formates) are most likely just spectators rather than active intermediates, because there are also weak formate species observed over ZnGaOx_F, which cannot be effectively converted by subsequent SAPO-34 catalysis to olefins". "Weak formate species observed over ZnGaOx_F, which cannot be effectively converted by subsequent SAPO-34 catalysis to olefins" seems to be correct because ZnGaOx_F is not effective in converting CO/H2 to methanol due to a low number of formates at the surface of ZnGaOx_F catalyst, but this does not make the formates spectators. To me formates are most likely a very important part of this catalyst system.

Line 140-146: The data are well explained by the selectivity to methanol, which have been demonstrated to be formed over both ZnGaOx_F and ZnGaOx_NP so an additional ketene mechanism is not needed to explain the observations.

Line 146-150: "The above results demonstrate that formation of ketene-acetate (acetyl) or formate species over ZnGaOx is critical for the reaction pathway towards final products, ..." This statement seems to be build upon two observations: (1) the yield over ZnGaOx/SAPO-34 mixtures and (2) some IR data where the sensitivity does not seem large enough for the ZnGaOx_F sample. Even if it could be demonstrated that no acetyl was observed at the surface of the ZnGaOx_F sample then the methanol yields in my opinion form a straightforward explanation strongly underpinned by literature.

Figure 4: Figure text. If the CO2 signal up to 400°C is potted in Figure 4b then these numbers should be available in Table S4, where the integrated CO2 signals in other temperature ranges are stated.

Why was H2 TPR not used instead of CO TPR?

Line 166-167: Integrated CO2 signals below 400°C are apparently not given in Table S4

Figure S7: The fitting procedures should be described. How was the background determined for example?

Line 174-175: Please indicate the position of the Mn impurity in Figure 4d.

Line 209-211: "This is further validated by AP-XPS analysis, which reveals a decreased O/Ga molar ratio of ZnGaOx-NP (the inset of Figure 5a)." The oxygen is bonded to both Zn and Ga and since Zn/Ga is reduced then the O/Ga should also be reduced. If this is so, then a lowering of the O/Ga ratio appears inconclusive.

Line 228: Fitting procedure including line shape should be given. It looks as if the fitting of the ZnGaOx_NP-A sample and the ZnGaOx_F samples in Figure S9(d,e) are not well-defined and that one could in another fit to the data in Figure S9(e) obtain more medium strength acid sites for the ZnGaOx-F sample.

Table S2: What are the methanol and the HC selectivities compared to each other? Please include these in the table.

Reviewer #3 (Remarks to the Author):

The bifunctional catalysis represented by OXZEO pioneered by the authors of this manuscript has made great breakthrough towards the direct conversion of CO to bulky feedstocks like ethylene and aromatics with high selectivity. Considerable research interests were previously devoted to tailoring the structures of oxide and zeolite components and the assemble manner. In this manuscript, the authors systematically compared and analyzed two different ZnGaOx spinel oxides and tested their direct syngas conversion to light olefines coupling with SAPO-34 zeolite component. They highlighted the importance of coordination unsaturated Ga³⁺ species and oxygen vacancies in oxide composite and achieved the steering of reaction pathway for syngas conversion. It is highly impressive that a high CO conversion (49.5%) is achieved over such bifunctional systems while keeping high selectivity to light olefins. The structures of both ZnGaOx spinel oxides were comprehensively characterized and the manuscript was well written. I recommend the acceptance of this work after the following issues are appropriately addressed.

1) Both hydrothermal and coprecipitation methods were employed to synthesize ZnGaOx samples with quite different structures and catalytic performance. Why the preparation methods lead to such striking difference?

2) From Table S3, it seems that both methanol and C2-C4 hydrocarbons could be produced by individual ZnGaOx nanoparticles. What is the reaction pathway for the production of these light hydrocarbons? Can the methanol-mediated pathway be completely excluded for the direct syngas conversion in bifunctional systems combing such ZnGaOx-NP and SAPO-34 composites? The authors proposed that the reaction occurs via the ketene-acetate (acetyl) intermediates by the coordination unsaturated Ga³⁺ and species and oxygen vacancies of ZnGaOx-NP/SAPO-34 systems.

3) The authors demonstrated that medium Lewis acid sites differentiate over two kinds of ZnGaOx oxides and analyzed by NH₃-TPD approach. While both Bronsted and Lewis acid sites can be characterized by such NH₃-TPD approach. Additional characterization methods may be required to analyze the distribution of surface acid sites.

4) How about the stability of the catalyst samples?

Reviewer #1 (Remarks to the Author):

Comment: Tandem catalysis is a key aspect in C1 chemistry and catalysis, for the conversion of CO, CO₂, methanol, methane, etc. Despite general recognition of the importance of defect sites (catalytic sites) on metal oxides including spinel in syngas (CO+H₂) conversion, there still exists less knowledge about the active sites on oxides for CO/H₂ activation and debate on the reaction routes because of its complexity.

This paper discloses deep insights on the defect sites of ZnGaO_x spinel. It is an interesting and timely study demonstrating the essential and intrinsic role of the expected types of defects, which can catalyze the reaction toward valuable chemicals such as olefins. The conclusion is convincing and well supported by multiple-aspect experimental evidences from a variety of characterizations including high resolution TEM, in situ FTIR, quasi-in-situ photoluminescence and EPR, besides AP-XPS, CO-TPR and NH₃-TPD. Therefore, I believe that this study deserves publication at Nature Communications. But I have several questions before acceptance.

Response: We appreciate the high opinions of the reviewer about this work. To address the reviewer's concerns in the following, we have carried out additional experiments.

Comment a): Concerning this bifunctional catalyst for a consecutive reaction, how is the effect of the relative ratio of metal oxides and zeolites? Is there any limitation at the activity of the second step, zeolite catalysis, for ZnGaO_x_F? I suggest the authors give more discussions on kinetics balance.

Response: We thank the reviewer for the insightful comment. It is indeed important to avoid the limitation at the activity of the second step of zeolite catalysis while studying the catalysis over metal oxides. Therefore, we carried out experiments by varying the zeolite content of the bifunctional OXZEO catalysts from 25 wt.% to 67 wt.%. The data were added in Fig. S8 of the revised Supplementary Information. The space time yield (STY) of light olefins for ZnGaO_x_F-SAPO-34 composites did not vary with the zeolite content, which indicates that the catalytic activity of ZnGaO_x_F was intrinsically low and there were not enough intermediates generated over ZnGaO_x_F. Therefore, the reaction equilibrium was not affected in the presence of much SAPO-34. There is not limitation at the activity of the second step of zeolite catalysis for ZnGaO_x_F under our reaction conditions.

By contrast, the yield of light olefins increases with the zeolite content in the bifunctional catalyst ZnGaO_x_NP-SAPO-34 and the one containing 50 wt.% zeolite gave a highest STY of light olefins. Therefore, there is also no limitation at the activity of

the second step for $\text{ZnGaO}_x\text{-NP}$ under our reaction conditions.

Fig. S8c. Comparison of STY of light olefins and total hydrocarbons as a function of zeolite content in the bifunctional catalysts $\text{ZnGaO}_x\text{-NP- SAPO-34}$ and $\text{ZnGaO}_x\text{-F- SAPO-34}$. Reaction conditions: $\text{H}_2/\text{CO} = 2.5$ (v/v), 400 °C, 4 MPa, 1,600 mL g⁻¹ h⁻¹.

Action taken:

We have added Fig. S8c and the above discussions in the revised Supplementary Information.

Comment b) More detailed analysis and interpretation of the EPR data, related to the reaction direction, is expected.

Response: In order to gain more intuitive and in-depth insights into the EPR results, we fitted the quasi-in-situ EPR spectra of the catalyst. The data are displayed in Fig. S13a in the revised Supplementary Information. Two species were presented on g-factors of 2.004 and 1.97 for the H₂-reduced $\text{ZnGaO}_x\text{-NP}$ sample. The species with g-factor of 2.004 were assigned to zinc vacancies (*Chemcatcher*, 2018, 10, 1536; *Phys. Lett. A*, 1970, 33, 1; *Sci. Sinter.*, 2004, 36, 65; *J. Mater. Sci.*, 1997, 32, 4619; *Semicond. Sci. Tech.*, 2020, 35, 095035) while those with g-factor of 1.97 was attributed to oxygen vacancies (*Appl. Phys. Lett.*, 2002, 81, 622; *Adv. Funct. Mater.*, 2005, 15, 1945).

Fig. S13a. Fitting curves of *quasi-in-situ* EPR spectra of the H₂-reduced ZnGaO_x_NP.

Action taken:

We have added Fig. S13a in the revised Supplementary Information.

Comment c) Figure S5 displays a new diffraction peak at 2theta 32 degree. But it is not discussed. Is this peak from ZnO? or from the interface of ZnO and ZnGa₂O₄ forming over the catalyst? Does it also contribute to the reaction?

Response: The diffraction peaks at 2-theta = ~ 32° and 34° correspond to (100) and (002) of ZnO impurity for ZnGaO_x_NP700 and ZnGaO_x_NP800 samples.

In order to judge the contribution of the few ZnO impurity, ZnGaO_x_NP700 was treated by dilute HNO₃, which removes the ZnO impurity (see XRD in Fig. S6a). The resulting catalyst does not show obvious change in the catalytic performance (Fig. S6b), indicating a margin effect of the ZnO impurity or the interface of ZnO/ZnGaO_x.

Fig. S6. Effect of few ZnO impurity on syngas conversion. a XRD patterns of ZnGaO_x_NP700

and ZnGaO_x_NP700-leaching samples. **b** Catalytic performances of ZnGaO_x_NP700–SAPO-34 and ZnGaO_x_NP700-leaching–SAPO-34 catalysts. Reaction conditions: oxide/zeolite mass ratio (OX/ZEO) = 1, H₂/CO = 2.5 (v/v), 400 °C, 4 MPa and 1,600 mL g⁻¹ h⁻¹.

Action taken:

We have added Fig. S6 and related discussion in the revised Supplementary Information.

Comment: d) References format need to be normalized.

Response: We have carefully checked all references and updated the format.

Reviewer #2 (Remarks to the Author):

Comment: “Steering the reaction pathway of syngas-to-light olefins with coordination unsaturated sites of ZnGaO_x spinel” by Na Li, Yifeng Zhu, Feng Jiao, Xiulian Pan, Qike Jiang, Jun Cai, Yifan Li, Wei Tong, Changqi Xu, Shengcheng Qu, Bing Bai, Dengyun Miao, Zhi Liu, and Xinhe Bao.

The main conclusions in the manuscript are:

- 1) That syngas conversion can be steered towards light olefins via ketene-acetate (acetyl) intermediates by the coordination unsaturated Ga^{3+} species and oxygen vacancies over ZnGaO_x spinel \square SAPO-34 composites.
- 2) Spinel containing only zinc vacancies leads to formation of formate intermediate species, but only as spectators, which can hardly be converted to light olefins by subsequent SAPO-34 catalysis.

As discussed below, the conclusion that the formate surface intermediates are not important does not appear to be well founded based on data. Formate species are generally believed to be precursors for methanol synthesis and there are significant differences between the two main samples (ZnGaO_{x_NP} and ZnGaO_{x_F}) of this study in the selectivity and activity to methanol. Methanol can subsequently be converted into olefins over SAPO-34. Consequently, the methanol selectivity and activity look as a straightforward explanation of the OXZEO data presented here. Furthermore, the quality of the manuscript could be improved by amending a number of imprecise statements/language, not sufficiently defined methods, considerations later turning into conclusions etc. In general I find that the manuscript does not contain the novelty and quality needed to merit publication in Nature Communication.

Response: We argue that this manuscript contains the novelty and quality, which merit publication in Nature Communications, from the following point of view.

First of all, this ZnGaO_{x_NP} gives a highest activity among the Cr-free metal oxides for syngas-to-light olefins. There are a number of metal oxides within the framework of OXZEO reported for syngas-to-light olefins. The highest conversion is obtained over Cr containing metal oxides (~50%) at an olefin selectivity of ~70-80%. Due to the concerns of the toxicity of Cr, there are wide efforts seeking highly active Cr-free metal oxides. In this work, ZnGaO_{x_NP} -SAPO-34 gives olefin selectivity as high as 75% at 49.5% CO conversion. This represents the highest yield among reported Cr-free metal oxide in the literature (**Table R1**). This is also acknowledged by the reviewer #3.

Table R1. Non-Cr-based metal oxides reported for syngas-to-light olefins.^a

Metal Oxide	CO conv. (%)	Light olefins sel. (%) ^b	Ref.
ZnZrO _x	7	69	Angew. Chem. Int. Edit. , 2016, 55 , 4725
MnO _x	7	79	ACS Catal. , 2017, 7 , 2800
ZnO	32	77	ACS Catal. , 2019, 9 , 960
Zr-doped In ₂ O ₃	28	74	ChemCatChem , 2018, 10 , 1536
MnGaO _x	14	88	Catal. Sci. Technol. , 2019, 9 , 5577
ZnGa ₂ O ₄	30	77	ACS Catal. , 2020, 10 , 8303
ZnAl ₂ O ₄	24	80	ACS Catal. , 2020, 10 , 8303
ZnGaO _x _NP	49.5	75	This work

^a SAPO-34 was used as zeotype component. ^b Selectivity in hydrocarbons. Note that CO₂ selectivity varied around 40-45%.

Secondly, we reveal here for the first time that not only just oxygen vacancies, but also the coordinatively unsaturated metal (Ga³⁺, in this case) sites, the B sites in ZnB₂O₄ spinel oxides, together with oxygen vacancies play a vital role in CO and H₂ activation. The in-depth insights gained here are also acknowledged by the reviewer #1. Furthermore, we show that this can be tuned by the morphology of ZnGaO_x oxide.

Therefore, we believe that this manuscript contains enough novelty and quality and the findings will attract wide attention in the fields of C1 chemistry, surface science, material science, theoretical calculations, and thus merits publication in Nature Communications.

Comment: Line 32-33: "SAPO-34 catalysis" I would say this is a phrase from a spoken language. Please re-phrase.

Response: We have deleted that sentence.

Comment: Line 46-49: Please also report conversion levels. This is very important information, when comparing selectivity of these reactions.

Response: We have added the conversion information as requested in the revised manuscript: “For example, the selectivity of light olefins among hydrocarbons reached more than 80% at 17% CO conversion over ZnCrO_x-SAPO-34 at 400 °C, 2.5 MPa³ while 49% CO conversion and 83% selectivity of light olefins over ZnCrO_x-AIPO-18 at 390 °C, 10 MPa⁸. ZnCrO_x-mordenite gave 83% ethylene selectivity and 7% CO conversion at 360 °C, 2.5 MPa⁶. ”

Comment: Line 51: “ZnGaO-SAPO-34” should it be “ZnGaO_x-SAPO-34”

Response: We apologize for the typo. We have corrected it in the revised manuscript.

Comment: Figure 1: Please provide scalebars for all parts of this figure. In the part of the figure text that covers (a,b), it is not stated what material the SEM images show.

Response: We apologize for the confusion. We have added clear scalebars and sample information in the revised Fig. 1 and its caption.

Comment: Line 83-86: The type of samples is not introduced before e.g. “Hydrothermal sample” is mentioned. The paper should contain a small intro paragraph with an introduction of the samples.

Response: We thank the reviewer for the advice. We have added a sentence introducing the samples, as requested: “ZnGaO_x oxides prepared by coprecipitation method were denoted as ZnGaO_x_NP and those by hydrothermal method were named as ZnGaO_x_F.”

Comment: Figure 2a: I presume that the XPS values represent measurements of the oxidic samples?

Response: The XPS values in Fig. 2a do represent measurements of the oxidic samples. This is made clearer in the caption of Fig. 2a and Table S1.

Comment: Line 109-111: “..than the flake one ...” represents spoken language and this sentence should be carefully rewritten. “ZnGaO_x_NP-B with a surface Zn/Ga

molar ratio of 1.2” In Table S1, the Zn/Ga molar ratio of this sample is 0.3.

Response: We thank the reviewer for this comment. We have rephrased the sentence: “all nanoparticle ZnGaO_{x-NP} samples demonstrate much superior performance than ZnGaO_{x-F} upon being physically mixed with SAPO-34 separately”.

We apologize for the mistake. It should be ZnGaO_{x-NP} with a Zn/Ga molar ratio of 1.2 instead of ZnGaO_{x-NP-B}. This is corrected in the revised manuscript.

Comment: Line 116-117: Even though this study is about using ZnGaO_x samples together with SAPO-34 for the OXZEO process, the purpose of this study is to unravel the different reactivities of dissimilar surface sites on the ZnGaO_x particles. This calls for a thorough investigation also of the catalytic properties of the ZnGaO_x samples themselves and this has only been done for two samples in this work (Table S3). This should be done for all ZnGaO_x samples studied here because these studies seem very informative as the products observed are not processed further on the active sites of SAPO-34 and in that way more complexed to analyze and the ZnGaO_x differences somewhat blurred. Taking the available activity data for the two tested pure ZnGaO_x samples, it seems to me that they explain the data with ZnGaO_x mixed with SAPO-34 well and provide an explanation that is different from that put forward by the authors: The ZnGaO_{x-NP} sample is quite selective towards methanol and my rough estimate suggests that methanol is (at least) in equilibrium (please provide information in Table S3 of how close the methanol is to equilibrium), while this is not the case for ZnGaO_{x-F}, which also forms more methane. Since methanol is at equilibrium for the ZnGaO_{x-NP} sample the selectivity towards methanol is underestimated if the data in Table S3 is used to estimate the selectivity to methanol directly. If this sample is mixed with SAPO-34, the methanol is converted to olefins, so the data in Table S3 seems to show that the difference between the ZnGaO_{x-F} and the ZnGaO_{x-NP} samples is just the methanol selectivity and activity. Please comment.

Line 119-120: “CO conversion is enhanced remarkably over ZnGaO_{x-NP} □ SAPO-34 (Figure S6a), indicating a shifted reaction equilibrium by the tandem catalysis of SAPO-34” This appears to be correct, but the data in Table S3 strongly suggests that it is the methanol, which is (known to be) converted to olefins over SAPO-34.

Line 121-123: “By contrast, introducing SAPO-34 to ZnGaO_{x-F} hardly affects the overall conversion (< 8%) and light olefin selectivity < 16% at a wide range of OX/ZEO ratios (Figure S6b).” This is not surprising considering the above analysis based on

methanol as the precursor for olefins.

Response: As suggested by the reviewer, we have tested the catalytic performance of all ZnGaO_x oxides. Considering the low activity of ZnGaO_x oxides alone as the catalyst for syngas conversion, GC data were analyzed using carbon normalization method. Furthermore, to get more reliable data with standard deviation values (errors), we synthesized and tested 3-5 batches for each ZnGaO_x catalyst. The data are displayed in Fig. S10, where the equilibrium methanol concentration calculated by HSC 9.0 is also displayed along, represented by the dashed line, as requested by the reviewer.

Fig. S10 shows that among the ZnGaO_x_NP catalysts, ZnGaO_x_NP, ZnGaO_x_NP700, and ZnGaO_x_NP800, give methanol concentrations around the thermodynamic equilibrium concentration of 0.29% while the other four oxides give lower methanol concentration. They are not correlated well with the hydrocarbon concentration obtained over OXZEO catalysts containing the corresponding oxide. Therefore, we believe that methanol is not the key intermediate for olefin synthesis on ZnGaO_x-SAPO-34, although its contribution cannot be excluded completely because SAPO-34 is a classical catalyst for methanol-to-olefins.

Fig. S10. Relationship between methanol concentration and hydrocarbon formation.

Methanol formation activity over ZnGaO_x oxides alone and hydrocarbon formation activity over ZnGaO_x_NP-SAPO-34 bifunctional catalysts were compared. Reaction conditions: H₂/CO = 2.5 (v/v), 400 °C, 4 MPa, 3,200 mL g⁻¹ h⁻¹.

Action taken:

Fig. S10 and the corresponding discussion are added in the revised manuscript.

Comment: Figure 3: It seems that the sensitivity of the In-situ FTIR is not large enough for the ZnGaO_{x_F} sample to judge whether the formate to acetyl ratio is the same as that for ZnGaO_{x_NP}.

Line 134-136: *“They (formates) are most likely just spectators rather than active intermediates, because there are also weak formate species observed over ZnGaO_{x_F}, which cannot be effectively converted by subsequent SAPO-34 catalysis to olefins”.*
“Weak formate species observed over ZnGaO_{x_F}, which cannot be effectively converted by subsequent SAPO-34 catalysis to olefins” seems to be correct because ZnGaO_{x_F} is not effective in converting CO/H₂ to methanol due to a low number of formates at the surface of ZnGaO_{x_F} catalyst, but this does not make the formates spectators. To me formates are most likely a very important part of this catalyst system.

Line 140-146: *The data are well explained by the selectivity to methanol, which have been demonstrated to be formed over both ZnGaO_{x_F} and ZnGaO_{x_NP} so an additional ketene mechanism is not needed to explain the observations.*

Line 146-150: *“The above results demonstrate that formation of ketene-acetate (acetyl) or formate species over ZnGaO_x is critical for the reaction pathway towards final products, ...” This statement seems to be build upon two observations: (1) the yield over ZnGaO_x/SAPO-34 mixtures and (2) some IR data where the sensitivity does not seem large enough for the ZnGaO_{x_F} sample. Even if it could be demonstrated that no acetyl was observed at the surface of the ZnGaO_{x_F} sample then the methanol yields in my opinion form a straightforward explanation strongly underpinned by literature.*

Response: To adequately address this concern, we conducted additional *in-situ* IR experiments. The results confirm no acetyl species on ZnGaO_{x_F}. Fig. S9a shows that both formate and acetate species are observed on ZnGaO_{x_NP600} and ZnGaO_{x_NP-A}. To further validate the important role of acetate, we compared the CO conversion of ZnGaO_x-SAPO-34 bifunctional catalysts and the intensity of surface acetate signal around 1525 cm⁻¹ (Fig. S9). It shows CO conversion correlates much better with the intensity of acetate species (Fig. 3b), whereas it does not correlate well with that of formate (Fig. S9b), indicating the crucial role of surface acetate species in syngas conversion. Since acetate species are essentially the adsorbed ketene, acetate/ketene could be active intermediates leading to olefins on ZnGaO_{x_NP}-SAPO-34 composites. Formation of acetate species over ZnGaO_{x_NP} was also validated recently by solid-state Nuclear Magnetic Resonance study, which can be further converted to ketene and

subsequently to olefins over zeolites (*Catal. Sci. Technol.*, 2022, **12**, 1289-1295).

Furthermore, formation of acetate species requires breaking of C-O bond and C-C coupling. However, this rarely occurs on typical methanol synthesis catalysts (including ZnO-ZrO₂, Cu-ZrO₂, and Cu-ZnO-Al₂O₃) and therefore acetate species were not reported previously on these catalysts (*Angew. Chem. Int. Ed.*, 2016, **55**, 4725; *ACS Catal.*, 2017, **7**, 8544; *Chem*, 2017, **3**, 323; *Nature Commun.*, 2019, **10**, 1166).

Fig. 3. Surface intermediates over ZnGaO_x oxides and the relationship with catalytic performance. **a** *In-situ* FT-IR differential spectra of syngas conversion over H₂-reduced ZnGaO_x_NP (navy line) and ZnGaO_x_F (brown line) at 400 °C. **b** CO conversion as a function of acetate intensity at 1525 cm⁻¹ of FT-IR spectra of different ZnGaO_x samples in Fig. S9a.

Fig. S9. *In-situ* FT-IR differential spectra of syngas conversion over H₂-reduced ZnGaO_x samples. **a** Samples of ZnGaO_x_NP, ZnGaO_x_NP600, ZnGaO_x_NP-A, and ZnGaO_x_F. **b** Relationship between formate intensity around 1589 cm⁻¹ and CO conversion.

Action taken:

We have added the catalytic performance of metal oxides and *in-situ* IR results in Fig. 3b, S9, and S10. The corresponding discussion has been added on pages 8 and 9 of the revised manuscript.

Comment: Figure 4: Figure text. If the CO₂ signal up to 400°C is potted in Figure 4b then these numbers should be available in Table S4, where the integrated CO₂ signals in other temperature ranges are stated.

Response and action taken:

Thanks for this suggestion. We have revised Table S4 (Table S5 in the revised version) as suggested.

Comment: Why was H₂ TPR not used instead of CO TPR?

Response: It was shown in our previous study that reduction of ZnCr₂O₄ (111) surface by CO is much more energy favored than by H₂ (*Science*, 2016, **351**, 1065). Therefore, we used CO-TPR instead of H₂-TPR since our purpose was only to compare the reducibility between different oxides.

Nevertheless, we carried out H₂-TPR experiments as suggested by the reviewer. The data are added in Fig. S12. An obvious H₂ consumption signal ($m/z = 2$) is observed for ZnGaO_x_NP below the reaction temperature of 400 °C, in contrast to a much less H₂ consumption for ZnGaO_x_F, which only takes place at above 400 °C. Therefore, H₂-TPR also reveals a much higher reducibility of ZnGaO_x_NP than that of ZnGaO_x_F below 400 °C, consistent with CO-TPR results in Fig. 4a.

Fig. S12. H₂-TPR profiles of ZnGaO_x samples. $m/z = 2$ (H₂) signals in the effluents were monitored by an online mass spectrometer.

Action taken:

H₂-TPR profiles are added in Fig. S12 of the revised Supplementary Information.

Comment: Line 166-167: Integrated CO₂ signals below 400°C are apparently not given in Table S4.

Response and action taken:

We have revised Table S4 (Table S5 of the revised version) by adding the integrated CO₂ signals below 400 °C.

Comment: Figure S7: The fitting procedures should be described. How was the background determined for example?

Response and action taken:

Shirley type background is used here for all samples. The detailed fitting parameters have been added in Table S4.

Table S4. The detailed fitting parameters of CO-TPR profiles in Fig. S11.^a

Sample	BG type and range (°C)	Peak 1					Area (a. u.)
		Position (°C)	FWHM (°C)	M	TS	TL	
ZnGaO _x _NP	Shirley 200-402	287	65	0	0.8	18	4860
ZnGaO _x _NP600	Shirley 180-380	287	61	0	0.8	18	3298
ZnGaO _x _NP-A	Shirley 256-513	381	108	0	1	0.01	1963

ZnGaO _{x_F}	Shirley	369	107	0	1	0.1	328
	260-570						
Peak 2							
ZnGaO _{x_NP}	Shirley	>400	--	--	--	--	0
	200-402						
ZnGaO _{x_NP600}	Shirley	>400	--	--	--	--	0
	180-380						
ZnGaO _{x_NP-A}	Shirley	438	51	45	1	3	6524
	256-513						
ZnGaO _{x_F}	Shirley	454	77	16	1	3	2576
	260-570						

^a Note: M = Gaussian-Lorentzian mixing (0 Gaussian: 1 Lorentzian), TS = asymmetry parameter, TL = asymmetry tailing parameter. FWHM: full width at half maximum. BG: background.

Comment: Line 174-175: Please indicate the position of the Mn impurity in Figure 4d.

Response and action taken:

As suggested, we have indicated the signal of Mn²⁺ impurity (six blue vertical lines) in Fig. S13b. This signal is similar to that widely reported for Mn-containing materials such as Mn-Doped YAlO₃ Single Crystals (*J. Phys. Chem. C*, 2022, **126**, 743-753).

Fig. S13b. Fitting curves of quasi-in-situ EPR spectra. EPR spectrum of ZnGaO_{x_F} in Fig. 4d, and the six blue vertical lines indicate the signal of Mn²⁺ impurity.

Comment: Line 209-211: “This is further validated by AP-XPS analysis, which reveals a decreased O/Ga molar ratio of ZnGaO_x-NP (the inset of Figure 5a).” The oxygen is bonded to both Zn and Ga and since Zn/Ga is reduced then the O/Ga should also be reduced. If this is so, then a lowering of the O/Ga ratio appears inconclusive.

Response and action taken:

As displayed in Scheme R1 below, just removing Zn atom but not the adjacent O will result in a decreased Zn/Ga ratio but not O/Ga (Case: 1). Only if both Zn and O are removed can lead to both reduced Zn/Ga and O/Ga at the same time (Case: 2). If only O is removed but not the neighbor Zn will only result in decreased O/Ga but not Zn/Ga (Case: 3). Thus, a lowering of the O/Ga ratio is conclusive.

Scheme R1. Schematic diagram of Zn or O removal from ZnGaO_x framework under different circumstances.

Comment: Line 228: Fitting procedure including line shape should be given. It looks as if the fitting of the ZnGaO_x-NP-A sample and the ZnGaO_x-F samples in Figure S9(d,e) are not well-defined and that one could in another fit to the data in Figure S9(e) obtain more medium strength acid sites for the ZnGaO_x-F sample.

Response and action taken:

As requested, we have added the fitting procedure in Table S7. The full width at half maximum, i.e., half-width (FWHM) and other parameters have been carefully

controlled to ensure a reliable fitting and they are consistent for all ZnGaO_x samples. In addition, we showed fitting of all NH₃-TPD profiles, especially for Fig. S9d and S9e (Figs. S16d and S16e in the revised version). Although there is some error, the trend is clear, consolidating the conclusion of this work. Taking ZnGaO_{x_F} as an example, whatever the fitting parameters change, the number of medium strength acids remains lower than 0.01. Therefore, we believe that the result given in Fig. 5d is comparable and acceptable. We have updated the fitting results in Figs. S16, 5d and Tables S7, S8 of the revised Manuscript and Supplementary Information.

Fig. S16. NH₃-TPD and its fitting curves. d ZnGaO_{x_NP-A}. **e** ZnGaO_{x_F}. The integral area of Peak 1 represents the amount of weak strength acid sites, and that of peak 2 represents the amount of medium strength acid sites. Table S8 lists the detailed integration parameters.

Fig. 5d. Mass specific hydrocarbon formation rate as a function of the amount of medium strength acid sites of ZnGaO_x estimated by NH₃-TPD. Reaction conditions: OX/ZEO = 1/4 (mass ratio),

$\text{H}_2/\text{CO} = 2.5$ (v/v), 400 °C, 4 MPa, 20,000 mL g⁻¹ h⁻¹.

Table S7. NH₃-TPD analysis of ZnGaO_x samples.

Oxide	Desorption amount of NH ₃ (mmol g ⁻¹)		
	Weak	Medium	Total
ZnGaO _x _NP	0.079	0.091	0.17
ZnGaO _x _NP600	0.068	0.082	0.15
ZnGaO _x _NP-A	0.028	0.052	0.13
ZnGaO _x _F	0.086	0.009	0.04

Table S8. The detailed fitting parameters of NH₃-TPD profiles in Fig. S16.^a

Sample	BG type and range (°C)	Peak 1: weak acid					Acid density (mmol g ⁻¹)
		Position (°C)	FWHM (°C)	M	TS	TL	
ZnGaO _x _NP	Linear 103-377	167	64	0	0.5	15	0.078
ZnGaO _x _NP600	Linear 103-384	165	63	0	0.5	10	0.069
ZnGaO _x _NP-A	Linear 104-380	166	65	0	0.5	30	0.028
ZnGaO _x _F	Linear 103-375	160	75	0	0.5	30	0.086
Peak 2: medium strength acid							
ZnGaO _x _NP	Linear 103-377	244	95	43	0.5	0.6	0.091
ZnGaO _x _NP600	Linear 103-384	238	94	55	0.5	0.6	0.082
ZnGaO _x _NP-A	Linear	244	95	50	0.5	0.6	0.052

	104-380						
ZnGaO _x F	Linear	244	90	50	0.5	0.6	0.009
	103-375						

^a Note: M = Gaussian-Lorentzian mixing (0 Gaussian: 1 Lorentzian), TS = asymmetry parameter, TL = asymmetry tailing parameter.

Comment: Table S2: What are the methanol and the HC selectivities compared to each other? Please include these in the table.

Response and action taken:

As requested, we have added the methanol selectivity in Table S2. Since HC selectivities can be obtained by deducing that of CO₂ and methanol, we did not add HC selectivities but gave hydrocarbon distribution in Table S2.

Reviewer #3 (Remarks to the Author):

Comment: *The bifunctional catalysis represented by OXZEO pioneered by the authors of this manuscript has made great breakthrough towards the direct conversion of CO to bulky feedstocks like ethylene and aromatics with high selectivity. Considerable research interests were previously devoted to tailoring the structures of oxide and zeolite components and the assemble manner. In this manuscript, the authors systematically compared and analyzed two different ZnGaO_x spinel oxides and tested their direct syngas conversion to light olefines coupling with SAPO-34 zeolite component. They highlighted the importance of coordination unsaturated Ga³⁺ species and oxygen vacancies in oxide composite and achieved the steering of reaction pathway for syngas conversion. It is highly impressive that a high CO conversion (49.5%) is achieved over such bifunctional systems while keeping high selectivity to light olefins. The structures of both ZnGaO_x spinel oxides were comprehensively characterized and the manuscript was well written. I recommend the acceptance of this work after the following issues are appropriately addressed.*

Response: We appreciate the reviewer's high opinions about our work.

Comment: *1) Both hydrothermal and coprecipitation methods were employed to synthesize ZnGaO_x samples with quite different structures and catalytic performance. Why the preparation methods lead to such striking difference?*

Response: The ZnGaO_{x_F} sample was prepared by hydrothermal method. Water and ethylenediamine were mixed as the structure directing agent (SDA). Use of ethylenediamine and other amine molecules has been demonstrated in synthesis of other materials with different morphologies. Our HRTEM images in Figs. 1c,d and S3a,b reveal that ZnGaO_{x_F} preferably exposes {111} crystal faces. Therefore, we presume that ethylenediamine preferentially adsorbs on {111} face of ZnGaO_x and thus inhibits the crystal growth along this direction during the crystallization process forming flake-like morphology.

In comparison, the ZnGaO_{x_NP} samples were prepared by coprecipitation method using metal nitrate aqueous solution and a mixed solution of sodium hydroxide and sodium carbonate as the precipitation agent, and no templates or SDA was used. This results in nanoparticles, which do not preferably expose specific crystal faces.

Comment: *2) From Table S3, it seems that both methanol and C2-C4 hydrocarbons*

could be produced by individual ZnGaO_x nanoparticles. What is the reaction pathway for the production of these light hydrocarbons? Can the methanol-mediated pathway be completely excluded for the direct syngas conversion in bifunctional systems combining such $\text{ZnGaO}_{x\text{-NP}}$ and SAPO-34 composites? The authors proposed that the reaction occurs via the ketene-acetate (acetyl) intermediates by the coordination unsaturated Ga^{3+} and species and oxygen vacancies of $\text{ZnGaO}_{x\text{-NP}}$ /SAPO-34 systems.

Response: We thank the review for the thoughtful comment. The reaction pathway for formation of $\text{C}_2\text{-C}_4$ hydrocarbons on individual ZnGaO_x nanoparticles was also suggested in our recent *quasi-in-situ* solid state NMR DFT study (*Catal. Sci. Technol.*, 2022, **12**, 1289-1295). Surface C1 species could react with CO to form ketene. In the absence of zeotypes, ketene was converted to thermodynamically stable acetate species and further to $\text{C}_2\text{-C}_4$ hydrocarbons, which was proved by NMR in previous work and also *in-situ* FT-IR in this work (Figure 3). Note that acetyl species are not a commonly observed on typical methanol synthesis catalysts (including ZnO-ZrO_2 , In_2O_3 , Cu-ZrO_2 , and $\text{Cu-ZnO-Al}_2\text{O}_3$) (*Nature Chem.*, 2017, **9**, 1019; *Angew. Chem. Int. Edit.*, 2017, **56**, 2318; *ACS Catal.*, 2011, **1**, 365; *Chem*, 2017, **3**, 323; *Nature Commun.*, 2019, **10**, 1166). However, they are clearly observed on $\text{ZnGaO}_{x\text{-NP}}$ catalysts by both *ss*NMR and *in-situ* FT-IR, indicating C-C coupling activity of $\text{ZnGaO}_{x\text{-NP}}$. Furthermore, Fig. S9 shows that CO conversion of ZnGaO_x -SAPO-34 bifunctional catalysts is well correlated with the IR intensity of surface acetate signal around 1525 cm^{-1} (Fig. S9, Fig. 3b). Therefore, surface acetate/ketene may be more active intermediates over these oxides.

To elucidate the role of methanol for olefin formation over $\text{ZnGaO}_{x\text{-NP}}$ -SAPO-34, we further conducted the catalytic performance test for all ZnGaO_x oxides without SAPO-34. Fig. S10 indicates that the methanol concentration over sole metal oxides does not correlate well with the corresponding hydrocarbon concentration in syngas conversion over the corresponding OXZEO catalysts. Therefore, methanol may not be a key intermediate for olefin synthesis over ZnGaO_x -SAPO-34. However, considering that SAPO-34 is a classical catalyst for methanol-to-olefins and there is also methanol formation over these oxides, we think that methanol contribution cannot be excluded completely over the corresponding bifunctional OXZEO catalysts.

Action taken:

We have added metal oxides performance and *in-situ* IR results in Figs. 3b, S9 and S10. The related discussion has been added on pages 8 and 9 of the revised Manuscript and Supplementary Information.

Comment: 3) The authors demonstrated that medium Lewis acid sites differentiate over two kinds of ZnGaO_x oxides and analyzed by NH₃-TPD approach. While both Brønsted and Lewis acid sites can be characterized by such NH₃-TPD approach. Additional characterization methods may be required to analyze the distribution of surface acid sites.

Response: Since pyridine (Py) is widely used as a probe molecule to distinguish the Brønsted and Lewis acid sites, we performed *in-situ* IR with Py adsorption on two kinds of ZnGaO_x oxides (ZnGaO_x_NP and ZnGaO_x_F). As displayed in Fig. S15, upon exposing ZnGaO_x_NP to Py (navy colored line), two bands assigned to Lewis acid sites appear at ~1450 and ~1605 cm⁻¹. However, no signal at ~1540 cm⁻¹ is observed, where the Brønsted acid sites generally respond. Therefore, there are only Lewis acid sites on ZnGaO_x_NP and it is reasonable to assign these NH₃-TPD signals to the Lewis acid sites. By contrast, the Lewis acid signal intensities are extremely low and also no Brønsted acid sites are observed for ZnGaO_x_F (wine line), which is consistent with the few medium strength Lewis acid sites reflected by NH₃-TPD in Fig. S11e.

Fig. S15. *In-situ* FT-IR differential spectra of pyridine adsorption over ZnGaO_x_NP (navy line) and ZnGaO_x_F (wine line).

Action taken:

We have added Fig. S15 in the revised Supplementary Information.

Comment: 4) How about the stability of the catalyst samples?

Response:

We have conducted a 120 h stability test on the bifunctional ZnGaO_x-NP-SAPO-34 catalyst. As shown in Fig. S7, the catalyst is rather good stable within 120 h. CO conversion remains 45.9% and light olefins selectivity 69.9%.

Fig. S7. Stability test of ZnGaO_x-NP-SAPO-34. A new batch of catalysts was used here.

Reaction condition: OX/ZEO = 1, H₂/CO = 2.5 (v/v), 400 °C, 4 MPa, and 1,600 mL g⁻¹ h⁻¹.

Action taken:

We have added Fig. S7 in the Supplementary Information and corresponding discussion on page 6 of the revised manuscript.

REVIEWERS' COMMENTS

Reviewer #1 (Remarks to the Author):

I read the revised manuscript and the Answer. It is acceptable with its present form as my questions are well answered, while corresponding actions are fabricated in the revised manuscript.

Reviewer #3 (Remarks to the Author):

I am satisfied with the revisions to my concerns, and suggest the acceptance of the revised version.

Point-by-Point Response to Reviewers' Comments

REVIEWERS' COMMENTS

Reviewer #1 (Remarks to the Author):

I read the revised manuscript and the Answer. It is acceptable with its present form as my questions are well answered, while corresponding actions are fabricated in the revised manuscript.

Response: We thank the reviewer for the comments.

Reviewer #3 (Remarks to the Author):

I am satisfied with the revisions to my concerns, and suggest the acceptance of the revised version.

Response: We thank the reviewer for the comments.